# MR Image Super Resolution By Combining Feature Disentanglement CNNs and Vision Transformers

**Dwarikanath Mahapatra**[1,2]                    DWARIKANATH.MAHAPATRA@INCEPTIONIAI.ORG
[1] *Inception Institute of AI, Abu Dhabi, UAE*
[2] *Faculty of Engineering, Monash University, Melbourne, Australia*

**Zongyuan Ge**[2,3,4]                                ZONGYUAN.GE@MONASH.EDU
[3] *Airdoc-Monash Research Australia*
[4] *Monash eResearch Centre Australia*

**Editors:** Under Review for MIDL 2022

## Abstract

State of the art magnetic resonance (MR) image super-resolution methods (ISR) using convolutional neural networks (CNNs) leverage limited contextual information due to the limited spatial coverage of CNNs. Vision transformers (ViT) learn better global context that is helpful in generating superior quality HR images. We combine local information of CNNs and global information from ViTs for image super resolution and output super resolved images that have superior quality than those produced by state of the art methods. We include extra constraints through multiple novel loss functions that preserve structure and texture information from the low resolution to high resolution images.

**Keywords:** MRI, super resolution, disentanglement, CNN, ViT

## 1. Introduction

Image super-resolution (ISR) takes low resolution (LR) image inputs and reconstructs its corresponding high resolution (HR) version thus enabling detailed examination of interesting regions. This is particularly relevant for medical image analysis where physics of the imaging systems limits the spatial resolution of radiological images (e.g. MRI, Xray) since obtaining HR images requires longer scanning time, and leads to lower signal-to-noise ratio and smaller spatial coverage (Plenge and et al, 2012). HR images provide more detailed information about local structures and textures resulting in higher accuracy in disease diagnosis and planning (Chen et al., 2018a). Since originally acquired LR images pose challenges for accurate analysis it is important to have a reliable ISR method.

Recent works demonstrate the potential of convolutional neural networks (CNNs) in generating HR images by using SRCNN (Dong et al., 2014, 2016), residual learning in VDSR (Very Deep Super Resolution) (Kim et al., 2016), and the information distillation network (IDN) (Hui et al., 2018). (Zhang et al., 2018) leverage hierarchical features in residual deep networks (RDN) while (Chen et al., 2018a) combine 3D dense networks and adversarial learning for MRI super resolution. MR images have inherent characteristics such as repeating structural patterns making them less complex than natural images. Secondly, they have a large proportion of background pixels. Since most approaches give the background and foreground equal importance it does not lead to good feature learning. Also, CNN methods

capture mostly local context information and do not explore the global aspects. Zhang et al. in (Zhang et al., 2021) propose a squeeze and excitation network to capture the global characteristics thus leading to improved super resolution output. However, squeeze and excitation relies on CNN features to capture global context which is not optimal.

Vision transformers (ViT) (Dosovitskiy et al., 2020) are an exciting new development that effectively capture long range contextual information from images. Given sufficient training data ViTs have been shown to outperform state of the art CNN based methods for classification and segmentation. In this work we propose to combine CNNs and ViT for performing super resolution of MR images. CNNs learn local details while ViT captures the global context much better than previously proposed methods. Inherent to the ViT is a self attention module that focuses on the important parts of the image and thus improves SR quality. Our method also uses feature disentanglement to improve super resolution.

## 2. Related Work

**MR Image Super-resolution:** ISR has been widely applied to MR images (Scherrer et al., 2012; Manjón et al., 2010), and spectroscopy MRI (Iqbal et al., 2019; Jain et al., 2016). Initial methods achieved multiple frame image super resolution via alignment of multiple noisy LR images which proved to be very challenging (Zhao et al., 2019). Recent deep learning based ISR approaches show superior performance for MR image super resolution (Chen et al., 2018b; Pham et al., 2017; Zhao et al., 2019) but use large models that pose challenges in real world settings. Zhang et al. in (Zhang et al., 2021) propose a squeeze and excitation attention network as part of a lightweight model for ISR. (Feng et al., 2021a) achieve multi contrast MRI super resolution using multi stage networks. (Hu et al., 2021) use graph convolution networks for MRI super resolution, while in other related work recent methods have proposed hybrid-fusion networks for Multi-modal synthesis of MRI (Zhou et al., 2020), and (Dar et al., 2019) synthesize multi-contrast MRI using conditional GANs.

**Attention Mechanism:** Attention mechanisms enables adaptive resource allocation by focusing on important image regions (Hu et al., 2017) and are popular for many tasks like image recognition (Ba et al., 2015) and image captioning (Xu et al., 2016), as well as ISR (Hu et al., 2018; Zhang et al., 2018). They can be highly effective for MRI super resolution due to repeating patterns of relatively simpler structures and less informative background.

**Vision Transformers:** Dosovitskiy et al. (Dosovitskiy et al., 2020) demonstrate state-of-the-art performance on image classification datasets using large-scale pre-training and fine-tuning, and (Carion et al., 2020; Zhu et al., 2021) use ViTs for object detection. Hierarchical vision transformers with varying resolutions and spatial embeddings (Liu et al., 2021; Wang et al., 2021a) have been used to reduce feature resolution, while (Esser et al., 2020) demonstrate success in high resolution image synthesis.Recent work on transformer-based models for 2D image segmentation include the SETR model that uses a pre-trained transformer encoder with different CNN decoders (Zheng et al., 2021) for multi-organ segmentation in (Chen et al., 2021a), and a transformer-based axial attention mechanism for 2D medical image segmentation (Valanarasu et al., 2021). Hatamizadeh et al. propose UNETR (Hatamizadeh et al., 2021) for 3D medical image segmentation using transformers as the main encoder of a segmentation network and directly connecting to the decoder via skip connections. For 3D medical image segmentation, (Xie et al., 2021) use a backbone

CNN for feature extraction, a transformer to process the encoded representation and a CNN decoder for predicting segmentation outputs, while (Wang et al., 2021b) use transformers in the bottleneck of a 3D encoder-decoder CNN for semantic brain tumor segmentation. However, none of the methods use ViT for medical image super resolution.

**Motivation And Contribution:** Context information is especially relevant for medical ISR since they provide additional cues to generate superior quality HR images. Our contributions are: 1) We combine CNNs and ViTs for image super resolution. Local contextual cues from CNNs and global information from ViTs result in superior quality super resolved images than those produced by state of the art methods. A pre-trained ViT is finetuned using self supervised learning. 2) Using multiple loss functions we incorporate extra constraints that preserve structural and semantic information in the generated super resolved image. 3) By comparing with results from (Hu et al., 2021) we also demonstrate our method's better ability to learn global features compared to graph based super resolution methods.

## 3. Method

### 3.1. Overview

Given a low resolution (LR) image $x \in \mathcal{R}^{N \times N}$ our objective is to train a model that outputs a high resolution (HR) image $y \in \mathcal{R}^{M \times M}$, where $M > N$. Figure 1 shows the workflow of our proposed method. The LR image $x$ goes through a generator network consisting of a series of convolution blocks and an upsampler that increases the image dimensions from $N \times N$ to $M \times M$. The discriminator module ensures $y$ satisfies the following constraints.

1. The HR and LR image should have similar semantic characteristics since a higher resolution version should not alter image semantics. For this purpose we disentangle the image into structure and and texture features, and ensure their respective semantic information is consistent across both images.

2. The HR image should preserve global and local context of the original LR image. To achieve it we use features extracted using a pre-trained ViT to effectively capture the relations in LR image and ensure this relationship is preserved in the HR image.

### 3.2. Vision Transformers

Vision transformers play an important role in our super resolution framework by serving as a robust and accurate feature extractor that integrates long range context and structural information. We use the ViT from UNETR (Hatamizadeh et al., 2021) pre-trained for MR image segmentation and fine tune it for our task. We briefly describe the architecture below (for full details please refer to Appendix B) and also explain our modifications. UNETR uses the contracting-expanding pattern consisting of a stack of transformers as the encoder which is connected to the decoder using skip connections. A 1D sequence from the 3D input volume $x \in \mathcal{R}^{H \times W \times D \times C}$ with image dimension $(H, W, D)$ and C input channels is created by dividing $x$ into $N = (H \times W \times D)/P^3$ flattened non-overlapping patches of size $P \times P \times P$ and denote this set as $x_v$. A linear layer projects the patches onto a $K$ dimensional embedding

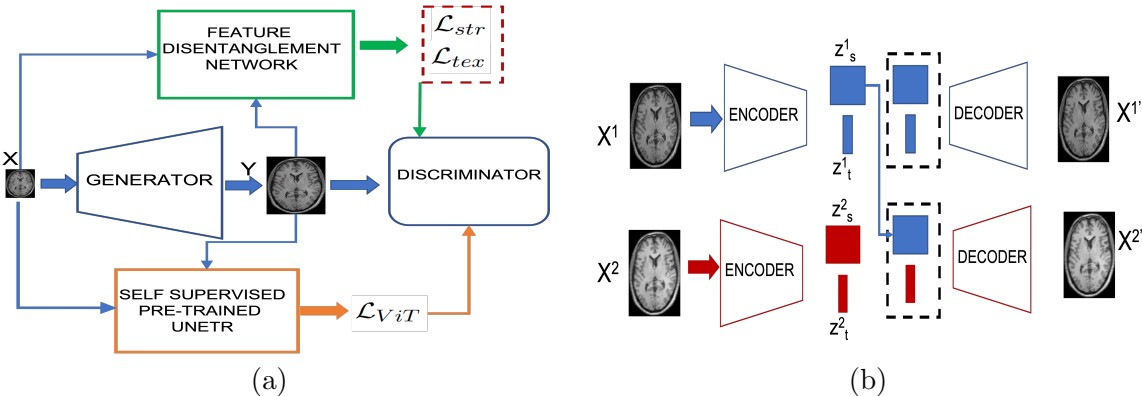

Figure 1: (a) Workflow of our proposed method. LR images goes through a generator to get HR image, and multiple loss functions ensure that semantic information of the LR image is preserved in the HR image. (b) Architecture of feature disentanglement network using swapped autoencoders.

space.To preserve spatial information a 1D learnable positional embedding $E_{pos} \in \mathcal{R}^{N \times K}$ is added to the projected patch embedding $E \in \mathcal{R}^{P^3.C \times K}$ as $z_0 = \left[ \mathbf{x}_v^1 \mathbf{E}; \mathbf{x}_v^2 \mathbf{E}; \cdots \mathbf{x}_v^N \mathbf{E} \right] + \mathbf{E}_{pos}$. Then multiple transformer blocks (Dosovitskiy et al., 2020) are used that have multi-head self-attention (MSA) and multilayer perceptron (MLP) sublayers according to

$$z_i' = MSA(\text{Norm}(z_{i-1})) + z_{i-1}, i = 1 \cdots L$$
$$z_i = MLP(\text{Norm}(z_i')) + z_i', i = 1 \cdots L \tag{1}$$

where $Norm()$ denotes layer normalization (Ba et al., 2016), MLP has two linear layers with GELU activation functions, $i$ denotes intermediate block and $L$ denotes transformer layers. $SA$ maps a query $(q)$ and the corresponding key $(k)$ and value $(v)$ representations in a sequence $\mathbf{z} \in \mathcal{R}^{N \times K}$. Attention weights $(A)$ measure similarity between elements in $z$ and their key-value pairs according to $A = \text{Softmax}\left( \frac{\mathbf{qk}^T}{\sqrt{K_h}} \right)$, where $K_h = K/n$ is a scaling factor. Thus, $SA(\mathbf{z}) = \mathbf{Av}$, where $v$ denotes input sequence values, and MSA output is:

$$MSA = [SA_1(\mathbf{z}); SA_2(\mathbf{z}); \cdots ; SA_n(\mathbf{z})] \mathbf{W}_{msa}, \tag{2}$$

where $\mathbf{W}_{msa} \in \mathcal{R}^{n.K_h \times K}$ represents the multi-headed trainable parameter weights.

**Self Supervised Learning:** A pre-trained transformer network such as UNETR has the advantage of being trained on medical images. We take the UNETR network and finetune it in a self supervised manner using images from the different datasets that we use for super resolution. Self supervised finetuning of ViT has attracted a fair bit of attention of late with different approaches using contrastive learning (Chen et al., 2021b) and masked auto-encoding (Chen et al., 2020; Dosovitskiy et al., 2020). We investigate both approaches and identify (Dosovitskiy et al., 2020) as more stable for our task. We remove the pre-trained prediction head and attach a zero-initialized $D \times K$ feedforward layer, where K is the number of downstream classes, and $D$ is the dimension of the flattened patches. We define a pre-text task to identify the primary organ in the images, which is akin to a classification problem involving $K$ classes.

### 3.3. Feature Disentanglement

In order to separate the images into structure and texture components we train an autoencoder (AE) shown in Figure 1 (b). In a classic AE the encoder $E$ and generator $G$ form a mapping between image $x$ and latent code $z$ using an image reconstruction loss

$$\mathcal{L}_{rec}(E, G) = \mathbb{E}_{x \sim X} \left[ \|x - G(E(x))\|_1 \right] \tag{3}$$

To ensure that the generated image is realistic we have discriminator $D$ that calculates the adversarial loss for generator $G$ and encoder $E$ as:

$$\mathcal{L}_{adv}(E, G, D) = \mathbb{E}_{x \sim X} \left[ -\log(D(G(E(x)))) \right] \tag{4}$$

As shown in Figure 1 (b) we divide the latent code into two components - a texture component $z_t$ and a structural component $z_s$. Then amongst similar images $X^1, X^2$ from the same dataset in a minibatch we swap the two components and enforce the constraint that the resulting images be realistic, using the 'swapped-GAN' loss (Park et al., 2020)

$$\mathcal{L}_{swap}(E, G, D) = \mathbb{E}_{x^1, x^2 \sim X, x^1 \neq x^2} \left[ -\log(D(G(z_s^1, z_t^2))) \right] \tag{5}$$

Here $z_s^1, z_t^2$ are the first and second components of $E(x^1)$ and $E(x^2)$. The intuition is to combine the structure component of one image with the texture component of another image. The two images are not identical although they belong to the same dataset. As shown in Figure 1 (b) the shapes of $z_s$ and $z_t$ are asymmetric. $z_s$ is designed to be a tensor with spatial dimensions so it can learn the structural properties associated with spatial configurations, and $z_t$ is vector that encodes the texture information. At each training iteration we randomly sample two images $x^1$ and $x^2$, and enforce $\mathcal{L}_{rec}, \mathcal{L}_{adv}$ for $x^1$, while applying $\mathcal{L}_{swap}$ to the combination of $x^1$ and $x^2$. The final loss function for *feature disentanglement* is given in Eqn. 6, and more details are given in Appendix A.

$$\mathcal{L}_{Disent} = \mathcal{L}_{Rec} + 0.7\mathcal{L}_{Adv} + 0.7\mathcal{L}_{swap} \tag{6}$$

We first train this disentanglement autoencoder that can extract the two separate features for a given input image (high or low resolution). The structure and texture features of the HR and LR images are used to train the super resolution network.

Since the HR and LR images are different versions of the same image swapping the structure code $z_s^{LR}$ (or texture $z_t^{LR}$) of the LR image with that of the HR image $z_s^{HR}$ (or $z_t^{HR}$) should still generate an image that is close to the original. Patches of size $n \times n$ are extracted around the center of the LR image and corresponding patches of size $mn \times mn$ are extracted from the center of the HR image, $m$ being the upscaling factor. This ensures that the two patches show the same region of interest. Swapping $z_t^{LR}$ with $z_t^{HR}$ and combining with $z_s^{LR}$ should produce an image very similar to the LR image. Similarly, $z_t^{LR}$ and $z_s^{HR}$ combine to give a fairly similar representation of the higher resolution image.

**Training The Super Resolution Network:** We use two pre-trained networks - the ViT and the feature disentanglement network. Given the LR image $x$ and the intermediate generated HR image $y$, we obtain their respective disentangled latent feature representations

as $z_s^x, z_t^x$ and $z_s^y, z_t^y$. Thereafter we calculate the semantic similarity between them using the cosine similarity loss as

$$\begin{aligned} \mathcal{L}_{str} &= 1 - \langle z_s^x, z_s^y \rangle \\ \mathcal{L}_{tex} &= 1 - \langle z_t^x, z_t^y \rangle. \end{aligned} \tag{7}$$

where $\langle . \rangle$ denotes cosine similarity. Additionally we also obtain the ViT based feature vectors of the HR ($f_{ViT}^{HR}$) and LR ($f_{ViT}^{LR}$) images from the ViT described previously and calculate their corresponding cosine similarity loss as

$$\mathcal{L}_{ViT} = 1 - \langle f_{ViT}^{LR}, f_{ViT}^{HR} \rangle \tag{8}$$

Once the above loss terms are obtained we train the whole super resolution network in an end to end manner using the following loss function. Thus the final loss function is

$$\mathcal{L}_{SR}(X,Y) = \mathcal{L}_{adv} + \lambda_1 \mathcal{L}_{ViT}(X,Y) + \lambda_2 \mathcal{L}_{str}(X,Y) + \lambda_3 \mathcal{L}_{tex}(X,Y). \tag{9}$$

## 4. Experiments And Results

**Dataset Description:** We use two datasets for our experiments: 1) **fastMRI** (Zbontar et al., 2019) - following (Xuan et al., 2020), we filter out 227 and 24 pairs of proton density (PD) and fat suppressed proton density weighted images (FS-PDWI) volumes for training and validation. 2) The **IXI dataset**: Three types of MR images are included in the datasets (i.e., PD, T1, and T2)[1]. Each of them has $500, 70$, and 6 MR volumes for training, testing, and validation respectively. Subvolumes of size $240 \times 240 \times 96$ are used and due to using 2D images we get $500 \times 96 = 48,000$ training samples.

| | IXI - PD Images | | IXI - T1 Images | |
|---|---|---|---|---|
| | 2× | 4× | 2× | 4× |
| | **PSNR/SSIM/NMSE** | **PSNR/SSIM/NMSE** | **PSNR/SSIM/NMSE** | **PSNR/SSIM/NMSE** |
| Bicubic | 30.4/0.9531/.042 | 29.13/0.8799/0.048 | 33.80/0.9525/0.030 | 28.28/0.8312/0.051 |
| (Feng et al., 2021b) | 31.7/ 0.892/ 0.035 | 29.5/ 0.870/ 0.033 | 30.7/ 0.883 /0.032 | 28.5/ 0.861/ 0.037 |
| (Dong et al., 2016) | 38.96 / 0.9836/0.022 | 31.10 / 0.9181/0.030 | 37.12 / 0.9761/ 0.26 | 29.90 / 0.8796/ 0.034 |
| (Zhang et al., 2018) | 40.31 / 0.9870 / 0.021 | 32.73 / 0.9387/ 0.029 | 37.95 / 0.9795/ 0.028 | 31.05 / 0.9042/ 0.031 |
| (Zhao et al., 2019) | 41.28 / 0.9895 / 0.02 | 33.40 / 0.9486/ 0.027 | 38.27 / 0.9810/ 0.025 | 31.23 / 0.9093/ 0.032 |
| (Zhang et al., 2021) | 41.66 /0.9902/0.019 | 33.97/0.9542/0.024 | 38.74/0.9824/0.021 | 32.03/0.9219/0.026 |
| (Hu et al., 2021) | 42.9/0.9936/0.018 | 35.3/0.962/0.023 | 39.9/0.989/0.021 | 33.6/0.927/0.024 |
| **Proposed** | 44.3/0.9972/0.016 | 37.1/0.972/0.021 | 41.4/0.993/0.019 | 35.4/0.9386/0.022 |
| **Ablation Studies** | | | | |
| $\mathcal{L}_{tex} + \mathcal{L}_{ViT}$ | 41.1/0.9826/0.021 | 35.3/0.958/0.025 | 39.5/0.983/0.021 | 33.1/0.924/0.024 |
| $\mathcal{L}_{str} + \mathcal{L}_{ViT}$ | 43.1/0.9902/0.018 | 35.8/0.963/0.023 | 40.5/0.986/0.021 | 34.5/0.9301/0.024 |
| $\mathcal{L}_{ViT}$ | 36.9/0.9745/0.027 | 34.2/0.943/0.027 | 37.2/0.962/0.025 | 31.3/0.903/0.028 |
| $\mathcal{L}_{tex} + \mathcal{L}_{str}$ | 37.6/0.9817/0.026 | 35.0/0.967/0.025 | 38.7/0.976/0.023 | 32.5/0.924/0.026 |

Table 1: Quantitative Results for IXI Dataset. Higher values of PSNR and SSIM, and lower value of NMSE indicate better results.

---

1. http://brain-development.org/ixi-dataset/

### 4.1. Implementation Details

**ViT Parameters**: For self supervised finetuning we use a batch size of 6 and cross entropy loss, the AdamW optimizer (Loshchilov and Hutter, 2019) with initial learning rate of 0.0001 for 20,000 iterations. For the specified batch size, the average training time was 10 hours for 20,000 iterations. **AE Network:** The encoder consists of 4 convolution blocks followed by max pooling after each step. The decoder is also symmetrically designed. $3 \times 3$ convolution filters are used and $64, 64, 32, 32$ filters are used in each conv layer. The input to the AE is $256 \times 256$ and dimension of $z_{tex}$ is 256, while $z_{str}$ is $64 \times 64$.

**Super Resolution Network:** We train our model using Adam (Kingma and Ba, 2014)) with $\beta_1 = 0.9, \beta_2 = 0.999$, a batch size of 256 and a weight decay of 0.1, for 100 epochs. We implement all models in PyTorch and train them using one NVIDIA Tesla V100 GPU with 32GB of memory. $\lambda_1 = \lambda_2 = 1$ and $\lambda_3 = 0.9$ (from Eqn.9).

### 4.2. Quantitative Results

For a given upscaling factor we first downsample the original image by that factor and recover the original size using different super resolution methods, and compare the performance using different metrics such as peak signal to noise ratio (PSNR), Structural Similarity Index Metric (SSIM), and Normalized Mean Square Error (NMSE). Tables 1, 2 show the average values of different methods for the IXI and fastMRI datasets at upscaling factors of 2× and 4×. Our method shows the best performance for both datasets and beats the next best method by a significant margin. While there is an expected noticeable performance drop for higher scaling factors, our method still outperforms other methods significantly. Our proposed method's advantage is the combination of CNN and ViT features that improve the image quality significantly. Although image quality degrades at higher magnification factor, our method performs better than others due to its ability to leverage local and global information.

**Ablation Studies:** Tables 1, 2 also show ablation study outcomes where different loss terms are excluded during training. Excluding the ViT features results in reduced performance. However it is still better than most other methods because of using feature disentanglement that leads to better super resolution based on texture and structure features. On the other hand excluding only one or more of structure and texture features leads to poor performance despite including ViT features. Thus we conclude that both global and local information is important for accurate super resolution.

### 4.3. Qualitative results:

In Figure 2 we show visualization results where the recovered images and their corresponding difference image with the original image is shown. Our method shows a very accurate reconstruction with minimal regions in the error map, while the recovered images from other methods are blurred and of poor quality. These results demonstrate the effectiveness of our approach.

|  | IXI-T2 Images | | | Fast MRI | |
|---|---|---|---|---|---|
|  | 2× | 4× |  | 2× | 4× |
|  | PSNR/SSIM/NMSE | PSNR/SSIM/NMSE |  | PSNR/SSIM/NMSE | PSNR/SSIM/NMSE |
| (Feng et al., 2021b) | 30.2/0.891/0.034 | 28.4/0.878/0.033 | (Lim et al., 2017) | 26.66/0.512/0.063 | 18.363/0.208/0.082 |
| (Dong et al., 2016) | 37.32/0.9796/0.027 | 29.69/0.9052/0.031 | (Zhao et al., 2018) | 28.27/0.667/0.051 | 21.81/0.476/0.067 |
| (Zhang et al., 2018) | 38.75/0.9838/0.026 | 31.45/0.9324/0.029 | (Lyu et al., 2020) | 28.870/0.670/.048 | 23.255/0.507/0.062 |
| (Zhao et al., 2019) | 39.71/0.9863/0.027 | 32.05/0.9413/0.031 | (Kim et al., 2016) | 29.484/0.682/0.049 | 28.219/0.574/0.059 |
| (Zhang et al., 2021) | 40.30/0.9874/0.022 | 32.62 / 0.9472/0.029 | (Feng et al., 2021a) | 31.769/0.709/0.045 | 29.819/0.601/0.054 |
| (Hu et al., 2021) | 41.9/0.991/0.020 | 34.2 / 0.951/0.027 | - | - | - |
| **Proposed** | 44.1/0.9953/0.017 | 35.4/0.959/0.024 | **Proposed** | 34.6/0.731/0.041 | 32.7/0.63/0.050 |
| **Ablation Studies** | | | | | |
| $\mathcal{L}_{tex} + \mathcal{L}_{ViT}$ | 40.8/0.977/0.021 | 33.6/0.941/0.027 | $\mathcal{L}_{tex} + \mathcal{L}_{ViT}$ | 32.1/0.713/0.046 | 30.4/0.60/0.054 |
| $\mathcal{L}_{str} + \mathcal{L}_{ViT}$ | 43.0/0.9875/0.020 | 34.3/0.947/0.026 | $\mathcal{L}_{str} + \mathcal{L}_{ViT}$ | 33.3/0.723/0.044 | 32.1/0.61/0.053 |
| $\mathcal{L}_{ViT}$ | 36.7/0.972/0.028 | 34.0/0.937/0.028 | $\mathcal{L}_{ViT}$ | 30.1/0.694/0.048 | 28.9/0.59/0.056 |
| $\mathcal{L}_{tex} + \mathcal{L}_{str}$ | 36.9/0.980/0.026 | 34.6/0.948/0.026 | $\mathcal{L}_{tex} + \mathcal{L}_{str}$ | 31.3/0.703/0.046 | 29.8/0.61/0.054 |

Table 2: Quantitative Results for IXI and fastMRI dataset super resolution output. Higher values of PSNR and SSIM, and lower value of NMSE indicate better results.

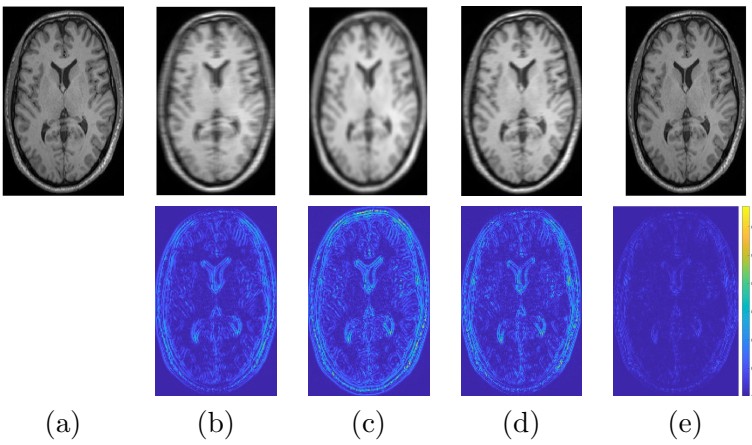

(a) (b) (c) (d) (e)

Figure 2: Visualization of superresolution results at 2× factor for the IXI dataset. The top row dhows the original image and the super resolved images and the bottom row shows the corresponding difference images.(a) Original image; Superesolved images obtianed using: b) (Zhang et al., 2021); (c) (Zhao et al., 2019); (d) (Zhang et al., 2018); (e) Our proposed method.

## 5. Conclusion

We proposed a novel method for MR image super resolution by combining CNNs and Vision transformers. ViTs provide more global context features while CNNs provide discriminative local information. We achieve feature disentanglement using swapped auto encoders to obtain texture and structure features. We enforce constraints that the original and super resolved images should have similar semantic information by minimizing the cosine loss of the respective structure and texture features, as well as minimizing the difference between the respective ViT features. Experimental results show our method outperforms state of the art techniques on benchmark public datasets, and ablation studies demonstrate the importance of our proposed loss terms.

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

## Appendix A. Feature Disentanglement

Similar to a classic autoencoder, the encoder $E$ produces a latent code $z \sim Z$ for image $x \sim X$. The $G$ reconstructs the original image from $z$ using an image reconstruction loss that is defined as:

$$\mathcal{L}_{Rec}(E, G) = \mathbb{E}_{x \sim X}\left[\|x - G(E(x))\|\right] \tag{10}$$

Additionally, the generated image should be realistic as determined by the Discriminator $D$ and is enforced using the adversarial loss defined as:

$$\mathcal{L}_{Adv}(E, G, D) = \mathbb{E}_{x \sim X}\left[-\log(D(G(E(x))))\right] \tag{11}$$

Furthermore, as part of our objective to achieve feature disentanglement we decompose the latent code $z$ into two components $[z_{str}, z_{tex}]$ corresponding to the structure and texture components. We enforce that swapping these components of the latent code with those from other images still produces realistic images. This is achieved by using a modified version of the adversarial loss, which we term as the swapped GAN loss, and is defined as :

$$\mathcal{L}_{swap}(E, G, D) = \mathbb{E}_{x^1, x^2 \sim X, x^1 \neq x^2}\left[-\log(D(G(z_{tex}^1, z_{str}^2)))\right] \tag{12}$$

Here $z_{tex}^1, z_{str}^2$ are the first and second components of images $X^1, X^2$'s latent representations, and $X^1, X^2$ from the same dataset in a minibatch. The component $z_{str}$ is a tensor with spatial dimensions, while $z_{tex}$ is a vector that encode structure and texture information. $\mathcal{L}_{Rec}$ and $\mathcal{L}_{Adv}$, are applied to image $X^1$ while $\mathcal{L}_{swap}$ is applied to the latent components from $X^1, X^2$. The final loss function for feature disentanglement is defined as

$$\mathcal{L}_{Disent} = \mathcal{L}_{Rec} + 0.7\mathcal{L}_{Adv} + 0.7\mathcal{L}_{swap} \tag{13}$$

## Appendix B. UNETR Architecture

We use the ViT from UNETR (Hatamizadeh et al., 2021) pre-trained for MR image segmentation and describe its architecture below. UNETR uses the contracting-expanding pattern consisting of a stack of transformers as the encoder which is connected to the decoder using skip connections. A 1D sequence of 3D input volume $x \in \mathcal{R}^{H \times W \times D \times C}$ with image dimension $(H, W, D)$ and C input channels is created by dividing it into flattened uniform non-overlapping patches $x_v \in \mathbb{R}^{N \times (P^3 \cdot C)}$ where $P \times P \times P$ denotes the resolution of each patch and $N = (H \times W \times D)/P^3$ is the length of the sequence.

A linear layer projects the patches onto a $K$ dimensional embedding space which remains constant throughout the transformer layers. To preserve spatial information a 1D learnable positional embedding $E_{pos} \in \mathcal{R}^{N \times K}$ is added to the projected patch embedding $E \in \mathcal{R}^{P^3 \cdot C \times K}$ according to

$$z_0 = \left[\mathbf{x}_v^1\mathbf{E}; \mathbf{x}_v^2\mathbf{E}; \cdots \mathbf{x}_v^N\mathbf{E}\right] + \mathbf{E}_{pos} \tag{14}$$

Then multiple transformer blocks (Dosovitskiy et al., 2020) are used that have multi-head self-attention (MSA) and multilayer perceptron (MLP) sublayers according to

$$z_i' = MSA(\text{Norm}(z_{i-1})) + z_{i-1}, i = 1 \cdots L \tag{15}$$

$$z_i = MLP(\text{Norm}(z_i')) + z_i', i = 1 \cdots L \tag{16}$$

where $Norm()$ denotes layer normalization (Ba et al., 2016), MLP has two linear layers with GELU activation functions, $i$ denotes intermediate block and $L$ denotes transformer layers. A MSA sublayer comprises of n parallel self-attention (SA) heads. Specifically, the SA block, is a parameterized function that maps a query $(q)$ and the corresponding key $(k)$ and value $(v)$ representations in a sequence $\mathbf{z} \in \mathcal{R}^{N \times K}$. Attention weights $(A)$ measure similarity between elements in $z$ and their key-value pairs according to

$$A = \text{Softmax}\left(\frac{\mathbf{q}\mathbf{k}^T}{\sqrt{K_h}}\right). \tag{17}$$

$K_h = K/n$ is a scaling factor for maintaining the number of parameters to a constant value with different values of the key $\mathbf{k}$. Using the computed attention weights, the output of SA for values v in the sequence z is computed as

$$SA(\mathbf{z}) = \mathbf{A}\mathbf{v}, \tag{18}$$

$v$ denotes input sequence values, and MSA output is:

$$MSA = [SA_1(\mathbf{z}); SA_2(\mathbf{z}); \cdots ; SA_n(\mathbf{z})]\,\mathbf{W}_{msa}, \tag{19}$$

where $\mathbf{W}_{msa} \in \mathcal{R}^{n.K_h \times K}$ represents the multi-headed trainable parameter weights.

At the encoder bottleneck (i.e. output of transformer's last layer), a deconvolutional layer is applied to the transformed feature map to increase its resolution by a factor of 2. The resized feature map is concatenated with the feature map of the previous transformer output and fed into consecutive $3 \times 3 \times 3$ convolutional layers, whose output is upsampled using a deconvolutional layer. This process is repeated for all the other subsequent layers up to the original input resolution where the final output is fed into a $1 \times 1 \times 1$ convolutional layer with a softmax activation function to generate voxel-wise semantic predictions.

## B.1. Loss Function

The loss function is a combination of soft dice loss and cross-entropy loss, and it can be computed in a voxel-wise manner according to

$$\mathcal{L}(G,Y) = 1 - \frac{2}{J}\sum_{j=1}^{J}\frac{\sum_{i=1}^{I} G_{i,j}Y_{i,j}}{\sum_{i=1}^{I} G_{i,j}^2 + \sum_{i=1}^{I} Y_{i,j}^2} - \frac{1}{I}\sum_{i=1}^{I}\sum_{j=1}^{J} G_{i,j}\log Y_{i,j} \tag{20}$$

where $I$ is the number of voxels; $J$ is the number of classes; $Y_{i,j}$ and $G_{i,j}$ denote the probability output and one-hot encoded ground truth for class $j$ at voxel $i$, respectively. For a detailed explanation of all terms we urge the reader to refer (Hatamizadeh et al., 2021).

The UNETR was implemented by the authors in PyTorch and MONAI and trained using a NVIDIA DGX-1 server. All models were trained with the batch size of 6, using the AdamW optimizer (Loshchilov and Hutter, 2019) with initial learning rate of 0.0001 for 20,000 iterations. For the specified batch size, the average training time was 10 hours for 20,000 iterations. The transformer-based encoder follows the ViT-B16 (Dosovitskiy et al.,

2020) architecture with L=12 layers, an embedding size of K=768. The patch resolution was $16 \times 16 \times 16$. For inference a sliding window was used with an overlap portion of 0.5 between the neighboring patches. The authors did not use any pre-trained weights for the transformer backbone (e.g. ViT on ImageNet) since it did not demonstrate any performance improvements for the medical images.

## Appendix C. Additional Visual Results

In this section we show additional visual results (Figures 3,4,5) from the IXI and Fast MRI dataset at different super resolution factors for the different ablation settings. The figures show the original image and the reconstructed image along with the difference image. They clearly illustrate the important contribution of each of the loss terms, and the adverse impact on super resolution if we exclude different terms.

## Appendix D. Computation Time

The original UNETR model has 92.58 Million parameters, and our finetuned model has similar number of parameters at 93.4 Million. The training time on a NVIDIA Tesla V100 GPU was 10 hours for $20,000$ iterations for the finetuning stage. The feature disentanglement network took 18 hours to train for 100 epochs. For the actual super resolution step, it took us 14 hours to train for 80 epochs. Note that feature disentanglement and ViT finetuning wer pre-trained and while training the super resolution network we only extracted features from them

The original UNETR model's inference time was 12.08s. Feature extraction from the finetuned UNETR model took 1.3s, while the disentangled feature extraction took 0.05 seconds per image. For the actual super resolution at inference stage it took 1.2 seconds for 2x upsampling for a $512 \times 512$ image

## Appendix E. Architecture of Super Resolution Network

Figure 6 shows the detailed architecture of the super resolution network's generator and discriminator components. In the generator (Figure 6 (a)) the input low resolution image $I^{LR}$ is passed through a convolution block followed by ReLU activation. The output is passed through a residual block with skip connections. Each block has convolutional layers with $3 \times 3$ filters and 64 feature maps, followed by batch normalization and ReLU activation. This output is subsequently passed through multiple residual blocks. Their output is passed through a series of upsampling stages, where each stage doubles the input image size. The output is passed through a convolution stage to get the super resolved image $I^{SR}$. Depending upon the desired scaling, the number of upsampling stages can be changed. The discriminator outputs the $\mathcal{L}_{adv}$ in Eqn 9, and is defined as:

$$\mathcal{L}_{adv,SR}(E, G, D) = \mathbb{E}_{lr \sim LR} \left[ -\log(D(G(E(lr)))) \right] \tag{21}$$

where $LR$ is the set of low resolution images and $G(E(lr))$ is the super resolved high resolution image. The other two loss terms, $\mathcal{L}_{ViT}, \mathcal{L}_{tex}, \mathcal{L}_{str}$, have been defined before.

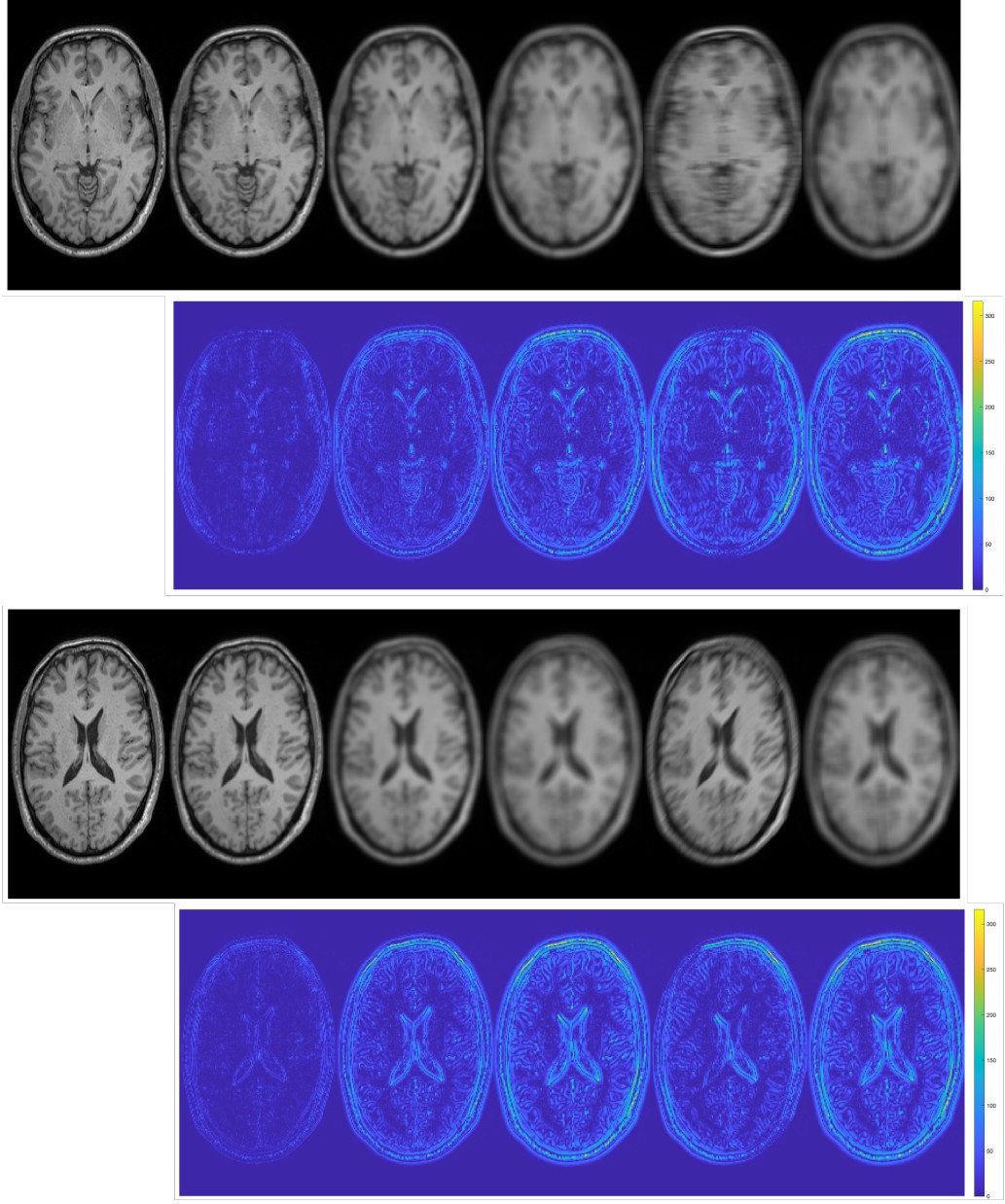

For IXI Image Dataset at 2x superresolution

Figure 3: Visualization of superresolution. For each figure the top row is the original image followed by the difference image inthe bottom row. Column 1- original image; Reconstructed Image using: Column 2- Our Proposed method; Column 3 - $\mathcal{L}_{str} + \mathcal{L}_{ViT}$; Column 4 - $\mathcal{L}_{tex} + \mathcal{L}_{ViT}$; Column 5 - $\mathcal{L}_{tex} + \mathcal{L}_{str}$; Column 6 - $\mathcal{L}_{ViT}$.

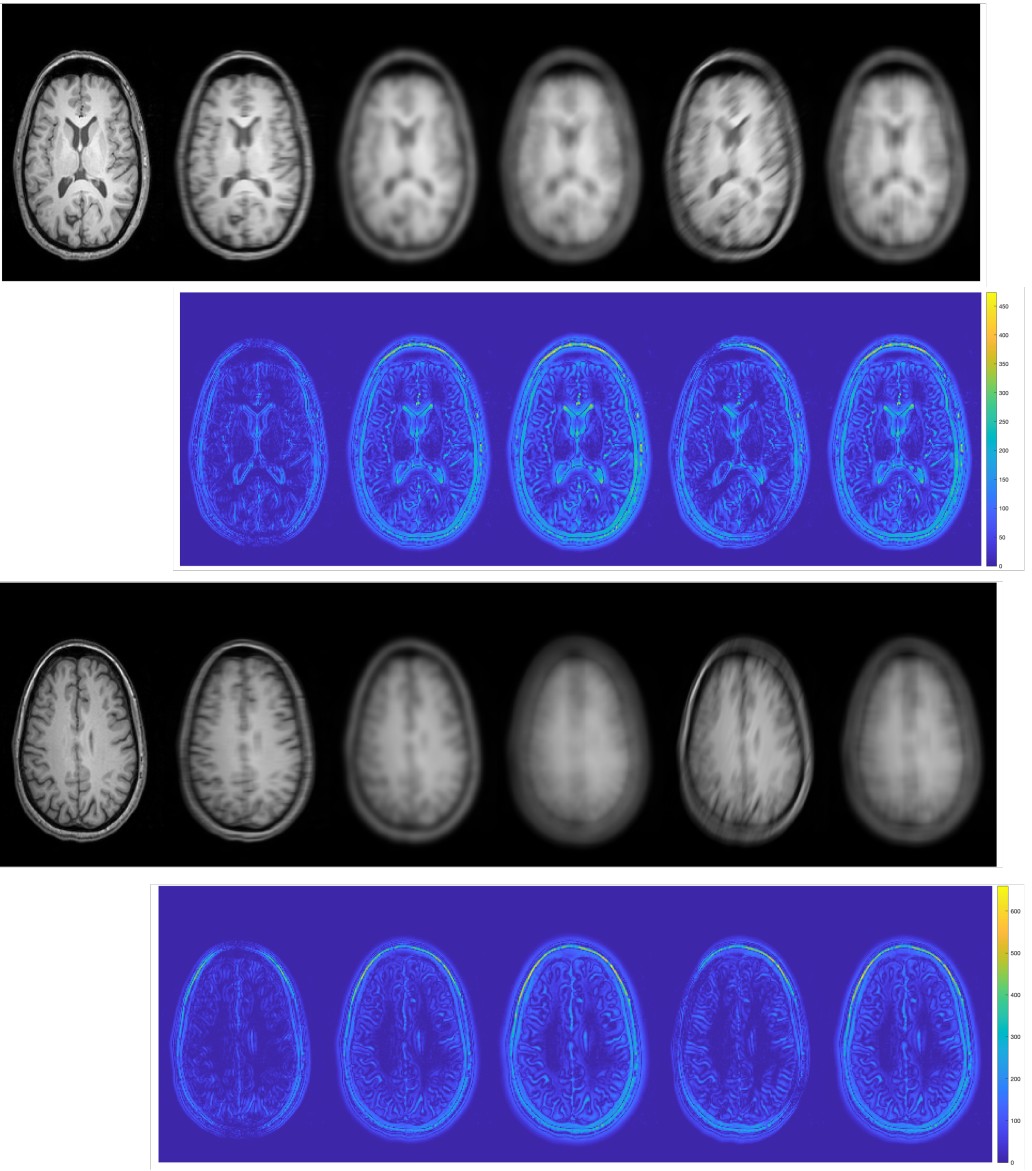

For IXI Image Dataset at 4x superresolution

Figure 4: Visualization of superresolution. For each figure the top row is the original image followed by the difference image inthe bottom row. Column 1- original image; Reconstructed Image using: Column 2- Our Proposed method; Column 3 - $\mathcal{L}_{str} + \mathcal{L}_{ViT}$; Column 4 - $\mathcal{L}_{tex} + \mathcal{L}_{ViT}$; Column 5 - $\mathcal{L}_{tex} + \mathcal{L}_{str}$; Column 6 - $\mathcal{L}_{ViT}$.

## Appendix F. Loss Plots

In figure 7 we show the loss plots for training, validation and test data splits ont he IXI brain image dataset. We see that the training error decreases gradually , which is also observable for the validation and test errors, although their magnitudes are higher than the

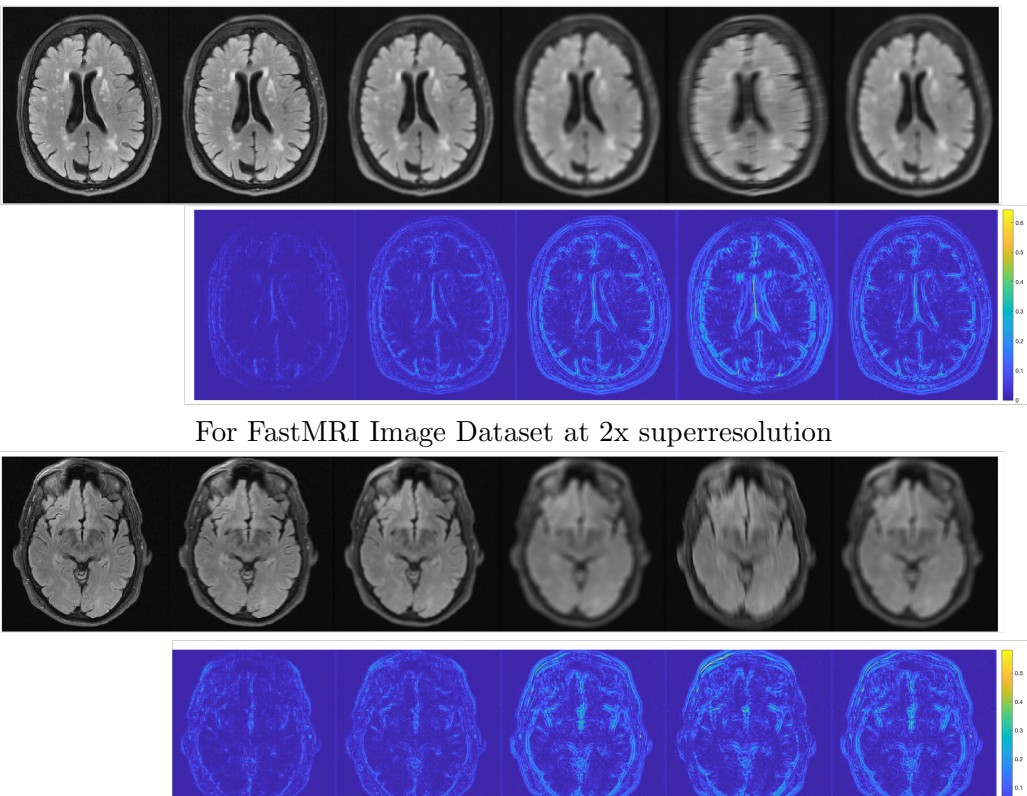

For FastMRI Image Dataset at 2x superresolution

For FastMRI Image Dataset at 4x superresolution

Figure 5: Visualization of superresolution results. For each figure the top row is the original image followed by the difference image inthe bottom row. Column 1- original image; Reconstructed Image using: Column 2- Our Proposed method; Column 3 - $\mathcal{L}_{str} + \mathcal{L}_{ViT}$; Column 4 - $\mathcal{L}_{tex} + \mathcal{L}_{ViT}$; Column 5 - $\mathcal{L}_{tex} + \mathcal{L}_{str}$; Column 6 - $\mathcal{L}_{ViT}$.

training error. The plots show that there is minimal chance of overfitting of the models and the results are not biased.

## Appendix G. Comparison With (Feng et al., 2021b)

The results of (Feng et al., 2021b) come up as worse than bicubic interpolation on the IXI dataset. This is surprising considering that they use a task transformer network. In our re-implementation we report better results than those reported on the paper (Feng et al., 2021b) since we devote significant bit of time in finetuning the parameters. Our experiments show that by removing the task transformer component the performance degrades but is still better than the numbers in (Feng et al., 2021b). While it is difficult to ascertain the reason behind their low performance, a possible reason could be the architecture of the task transformer network. This requires further investigation and is beyond the scope of our current work.

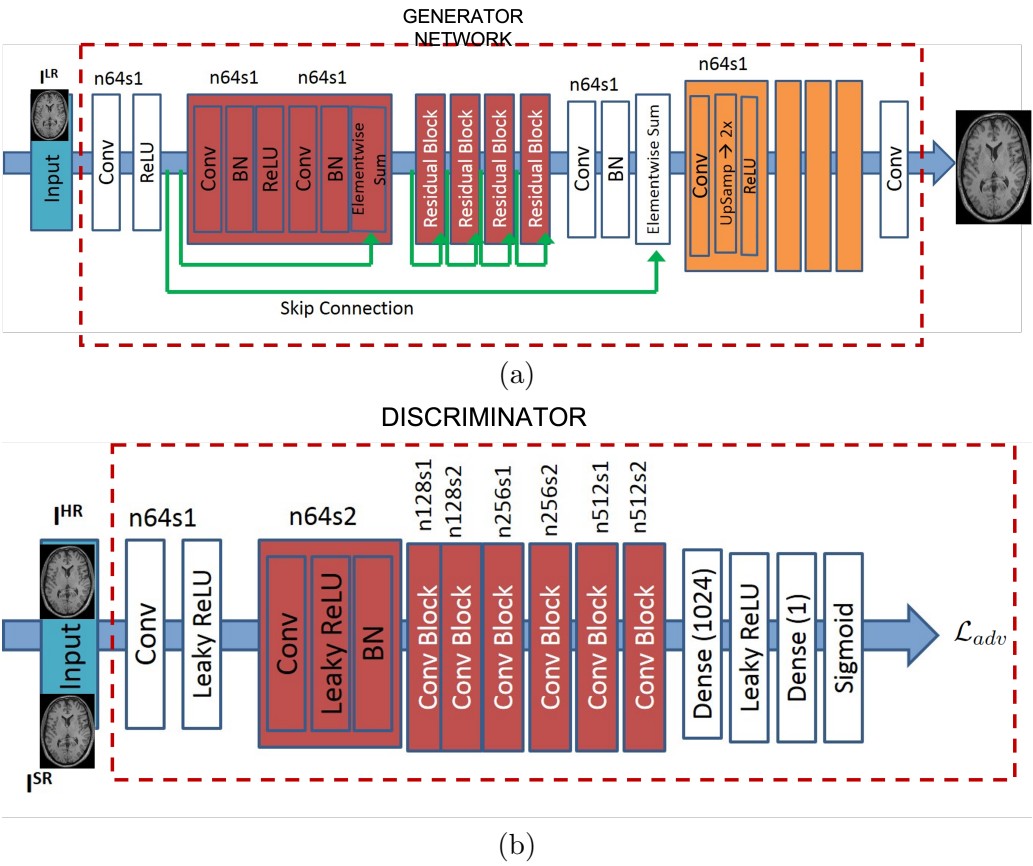

Figure 6: (a) Generator Network; (b) Discriminator network. $n64s1$ denotes 64 feature maps (n) and stride (s) 1 for each convolutional layer..

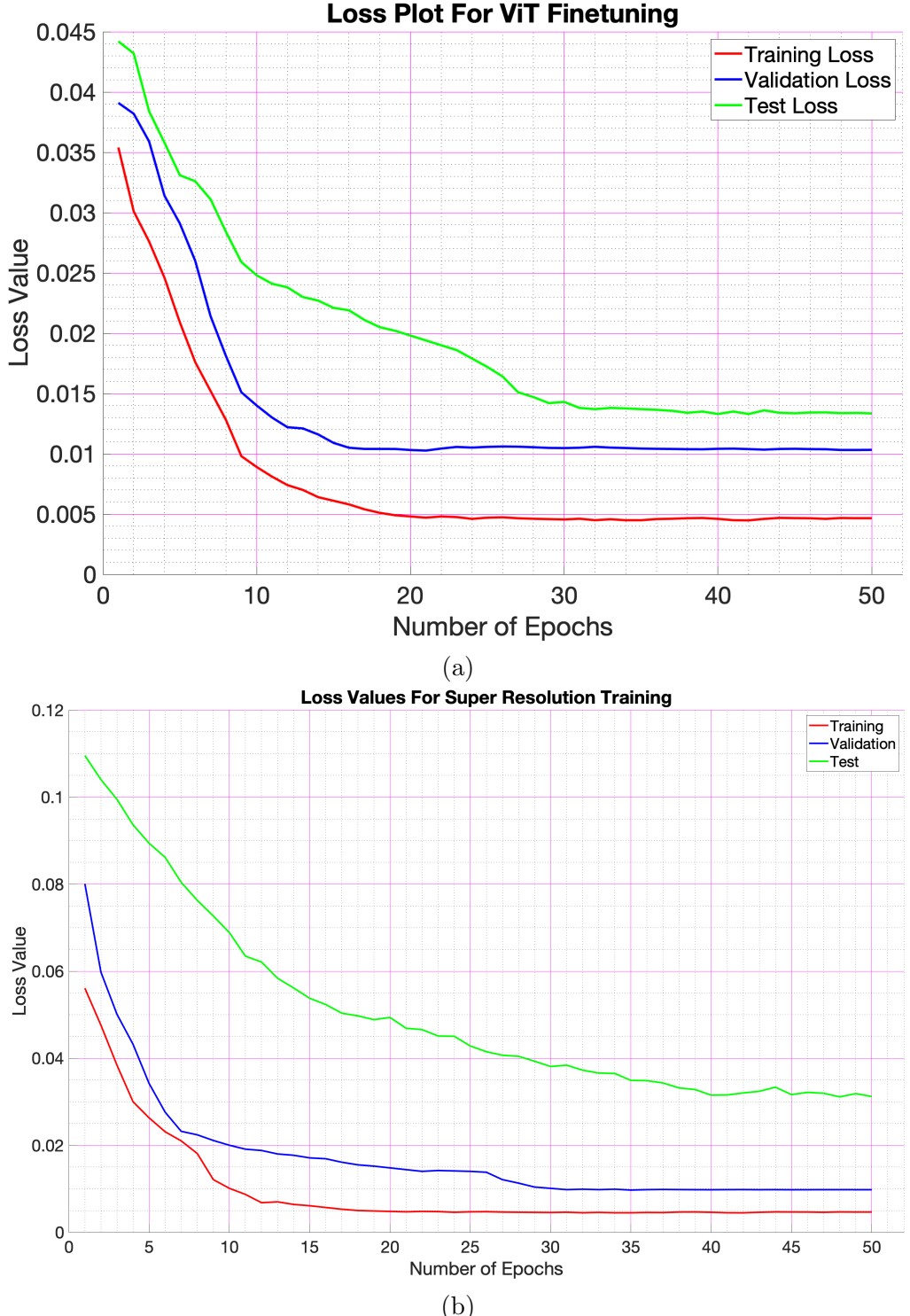

(a)

(b)

Figure 7: Loss plots for (a) UNETR Fine tuning using the IXI dataset; (b) Image super-resolution training for IXI dataset.

