# OpenReview forum: "MR Image Super Resolution By Combining Feature Disentanglement CNNs and Vision Transformers"
_MIDL.io/2022/Conference — MIDL 2022_

### Official Review · Reviewer_ytY8 · 2022-01-22

**Confidence:** 4
**Preliminary Rating:** 4
**Recommendation:** Poster

**Summary:**

The authors propose to inherit the complementary benefits of combining local information of CNNs and global information from ViTs for MR image super resolution.

The LR image is passed through a generator to generate an HR image.
The LR image and the HR output is passed through both the pretrained Vision transformer and the Variational autoencoder (feature disentanglement network - FDN)
The Pretrained vision transformer classifies the primary region in both LR input and HR output and it includes a cosine loss between them
The FDN gives out content and texture features for both LR and HR output. Model includes a cosine loss term for both the features
The FDN is pre-trained by selecting 2 images at random and while using the second image texture feature is replaced and swap loss is included.
Finally a discriminator is included in the model to ensure the generated image is realistic.


**Strengths:**

The work considers preservation of information at different levels of abstraction like semantic information, global and local context, detail (structure and texture) recovery and self-similarity. For an image restoration operator like super-resolution, inheriting these benefits is important.

The authors have taken an emerging trend of using vision transformers from computer vision based deep learning.

The authors propose to combine the structure component of one image with the texture component of another image, which is quite interesting.

**Weaknesses:**

Section 3.3 -> should still lead to accurate reconstruction. ----> This seems to be a strong claim without citation or adequate reasoning. The reasoning behind the assistance should be explained clearly. The intuition behind including the swap loss is not clear. In the original paper they swap the texture to actually replace the texture of the original image.

Equation 8, the image reconstruction loss component is missing. Does it mean that it is not used? There is a separate generator in the model too.

It is not clear if VAE or AE is used for the concept mentioned in the paper. There is no constraint in the latent space. The base paper is mentioning it as autoencoder https://arxiv.org/pdf/2007.00653.pdf

**Deanonymize Review:**

no

**Detailed Comments:**

outptut -> typo error the introduction

Section 3.3 -> anda

The notations for z_{s} and z{t} are missing in Figure 1b. Instead different notations are provided which affects readability of the paper.

In figure 1b, there is a character x between the red and blue portions, its not clear what this means. Similarly there is another character below the green box.

In the table, the entry for the method proposed by authors should be "Ours" or "Proposed method". In both tables left column has no heading.

**Final Rating After The Rebuttal:**

4: Weak Accept

**Justification Of The Final Rating:**

The rebuttal went on well. The authors were given inputs to improve the readability of the manuscript with respect to methodology, figures and  state-of-the-art comparison in the experiment section. The authors have followed the suggestions and made changes accordingly.
Numerical results on different MR datasets confirm its competitiveness well. Hence I would stick to my initial rating.

**Paper Type:**

methodological development

**Questions To Address In The Rebuttal:**

The authors need to specify the information about the image reconstruction loss component.

Adequate reasoning for using a VAE over AE must be provided.

Why bicubic interpolation is not shown in the tables? The reason for this question is that the method by (Feng et al., 2021b) is a 2021 paper but seems to perform poor. Adequate reasoning for such a performance drop is missing in Section 4.2. Bicubic interpolation will assess the relative performance of the models.



**Special Issue:**

no

---

### Official Review · Reviewer_S9zb · 2022-01-24

**Confidence:** 3
**Preliminary Rating:** 4
**Recommendation:** Poster

**Summary:**

The authors proposed an MRI super-resolution pipeline incorporating the vision transformer and features disentangle techniques. As a part of the discriminator, the transformer is employed to extract the long-range global features while the feature disentangle aims to pose the constraints on sematic and texture aspects. The experiments show that the pipeline achieves better performance than other most recent baselines.

**Strengths:**

The experiments show its better quantitative and qualitative performance than other baselines for MRI SR tasks. Ablation studies also demonstrate the effectiveness of each component in network. Paper is well-organized and easy to follow for readers.

**Weaknesses:**

Some important works are missed.
[1].Feedback Graph Attention Convolutional Network for MR Images Enhancement by Exploring Self-Similarity Features
[2]. Hi-net: hybrid-fusion network for multi-modal mr image synthesis
[3]. Image synthesis in multi-contrast mri with conditional generative adversarial networks.

Though the transformer has a powerful ability to extract global and long-range knowledge, it suffers from memory and computational cost. It is better to provide the time and GPU memory of the network both in the training and test stage.

Please provide the details of the generator and discriminator structure which is easy to be reimplemented.

The transformer backbone of Vit is used here. How about the performance of other transformer backbones such as swin transformer[4]?
[4].Hierarchical Vision Transformer using Shifted Windows


**Deanonymize Review:**

no

**Final Rating After The Rebuttal:**

4: Weak Accept

**Justification Of The Final Rating:**

After reviewing all rebuttal material and discussion, I am happy to keep my rate to weak accept.  Thanks for the authors' clarification to dismiss some questions.  This work probably inspires some research in the domain of medical image synthesis.

**Paper Type:**

both

**Questions To Address In The Rebuttal:**

To emphasize the novelty of the network rather than only using the existing network structure (transformer and features disentangle ); To analyze the inference time and computation cost; To discuss more recent relevant works.

**Special Issue:**

no

---

### Official Review · Reviewer_cThc · 2022-01-24

**Confidence:** 2
**Preliminary Rating:** 3
**Recommendation:** Poster

**Summary:**

This work proposes a method to compute super-resolved MRI images using a combination of generative CNN models, vision transformers (ViT) with an adapted loss function encouring feature disentanglement (structure and texture).

The method is tested on the fastMRI and IXI datasets and compared quantitatively with five different superresolution methods and with different combinations of the proposed loss functions.
Also a few qualitative results are shown.
According to these experiments, the proposed method outperforms the other approaches by a considerable margin.

**Strengths:**

The paper presents an interesting combination of several advanced approaches (e.g. vision transformers, self-supervised training, perceptual losses) to tackle MR superresolution. The quantitative results are very good.

**Weaknesses:**

The methods are not very well explained in this paper, and this makes it very hard to judge the results for me. This might be an effect of the limited size for a conference paper, but in this case the authors might consider writing a longer, more detailed paper for a journal. In its current form, the paper does not allow the reader to follow the methodological details in my opinion.

The paper should also provide more qualitative (visual) results to e.g. judge the effect of the loss ablations.



**Deanonymize Review:**

no

**Detailed Comments:**

I don't have enough expertise in the field of ViT, but I am a bit puzzled by the quality of the results. From figure 2 it seems like the method is able to reconstruct details that are at the resolution of single pixels and even some noisy texture patterns. Could this be an effect of memorization or overfitting? Can you rule out that this is happening?

How many parameters does the ViT model have? What was it trained on?

The MR images in Fig.2 have the wrong aspect ratio. I assume this is only by accident, but it should be corrected.

Would it be possible to publish the code? I think this would be important.

**Final Rating After The Rebuttal:**

3: Borderline

**Justification Of The Final Rating:**

The authors have added some more information and figures as a supplement. I think this might help the reader grasp more of the details.  Also, some of the results in the appendix look a bit more realistic to me.
However, overall I still find the main text hard to follow and I am not entirely convinced it fits such a short proceedings format.

**Paper Type:**

methodological development

**Questions To Address In The Rebuttal:**

The methods should be explained more clearly. If the format is too short, maybe there is room to cut some of the related work part.

The code should be published if possible.

It would be important to also see some qualitative results of the ablation study, the 4x undersampling and the fastMRI data, e.g. in a supplement or online repository.



**Special Issue:**

no

---

### Meta-Review · Area_Chair_ftrW · 2022-02-19

**Recommendation:** Accept (Poster)
**Confidence:** 5

**Metareview:**

The authors have addressed all the points raised by the Reviewers to some extent, including more experiments and discussion points to support the described method. AnonReviewer3 indicated that "this is a strange observation. But I think the discussion points for consistent reduction in the method by Feng et al needs to be highlighted somehow.". I do agree that it would need further clarification and discussion in the final camera-ready version. However, all Reviewers agree that the paper represents a contribution as a methodology paper and, considering the rebuttal effort, I recommend accepting this work for publication.

---

### Decision · Program_Chairs · 2022-02-28

Accept